# Treating the disease and meeting the person with the illness-patient perspectives of needs during infective endocarditis, a qualitative study

Helena Lindberg[1,2]* , Johan Vaktnäs[3‡], Magnus Rasmussen[2‡], Ingrid Larsson[4,5,6]

**1** Department of Infectious Diseases, Hospital of Halland, Halmstad, Sweden, **2** Department of Clinical Sciences Lund, Division of Infection Medicine, Lund University, Lund, Sweden, **3** Department of Oncology and Palliative Medicine, Hospital of Halland, Varberg, Sweden, **4** Department of Health and Nursing, School of Health and Welfare, Halmstad University, Halmstad, Sweden, **5** Spenshult Research and Development Centre, Halmstad, Sweden, **6** Department of Clinical Sciences, Section of Rheumatology, Lund University, Lund, Sweden

☯ These authors contributed equally to this work.
‡ JV and MR also contributed equally to this work.
* helena.lindberg@regionhalland.se

**Data Availability Statement:** The datasets analysed during the current study are not publicly available due to the integrity of the interviewees but are available from the corresponding author or the

## Abstract

### Background

Infective endocarditis (IE) is a rare but severe infectious disease. Patients with IE are treated for weeks in the hospital and have profound impairments to their health. New treatment modalities increase options for outpatient care. Little is known about how patients perceive their disease and hospitalisation. We aimed to explore the needs of patients with IE during hospitalisation and the first few months after discharge.

### Methods

In this qualitative study, 20 patients (45–86 years of age) hospitalised due to IE in Swedish hospitals were interviewed a median of 112 (67–221) days after hospitalisation. Data were analysed with qualitative content analysis, identifying eight subcategories, two categories, and an overall theme.

### Results

The overall theme illuminated a spectrum of needs of patients suffering from IE, between treating the disease and meeting the person with the illness. The needs encompassed eight axes with dual focus on both medical excellence and person-centred care. Medical excellence was needed to optimally treat, supervise, and offer follow-up on this rare and severe disease; patients longed to come home, and there were issues of reliability in the healthcare system. Person-centred care was requested, including individualised information leading to knowledge, reorientation, the beginning of health restoration, and being met as a unique

data access committee of Region Halland (dataskydd@regionhalland.se) upon reasonable request.

**Funding:** The author(s) received no specific funding for this work.

**Competing interests:** The authors have declared that no competing interests exists.

person. Symptoms of fatigue, wasting, and cognitive and mental distress were often neglected by the caregiver.

## Conclusions

This explorative study shows the patient's needs as important areas in a spectrum between medical excellence and person-centred care. Care in specialised units secure quality. Early discharge is requested by patients. Multiprofessional individualizing outpatient care needs to develop with preserved safety and medical excellence. The disease trajectory after discharge progresses slowly, and the possibility of mitigating its progress is still unclear. Person-centred care, screening for delayed restoration and rehabilitation after endocarditis are important fields for future studies.

## Introduction

Infective endocarditis (IE) affects the cardiac valves [1] of 6–8/100,000 persons in Sweden yearly [2–5]. In-hospital mortality is 10%–17% [2, 6], but 1-year mortality is up to 40% [6, 7], which is higher than mortality in most other types of cardiac disease [8]. The clinical picture of IE is highly variable, ranging from severe sepsis and embolic stroke to longstanding fever, night sweating, fatigue, and weight loss [5, 9]. Valvular disease, implanted cardiac devices, intravenous drug use, and old age increases the risk of IE [6, 10, 11]. Healthcare costs due to hospitalisation are high [12], with raised secondary costs during the postinfection period [8, 13]. Cardiac surgery improves the prognosis is performed in around 30% of cases, but has inherent risks [1, 5, 14, 15].

Mortality and readmission rates are high after IE, and self-reported physical and mental health is poor [8]. Patient perspectives in IE has indicated posttraumatic stress syndrome, increased anxiety and depression [16–18].

Treatment of IE typically means several weeks of hospitalisation with high doses of intravenous antibiotics [1, 6] and around 30% of patients experience adverse reactions [1, 6, 15]. The treatment of IE is facing a paradigm shift [19], with increasing possibilities for outpatient treatment either intravenously [20] or orally [21]. An important component of effective illness management is person-centred care, focusing on the needs, preferences, and values of patients. Person-centred care is an ethical standpoint guiding practical actions where partnership between patient and caregiver is a central aspect. Time spent meeting with patients as active and responsible partners, listening to their narrative, and forming personal health plans safeguards person-centredness and the involvement of multiple professions can improve and strengthen postadmission care [22, 23]. This repays in shorter time of care and higher patient satisfaction [24].

IE has thus an increasingly high impact on health care [19, 25–27], and changes of treatment are at hand. How patients' lives are impacted in the short and long term is not well known and knowledge of patients' experiences in the current healthcare setting can point to important areas for improvement and contribute to adapting new models of care that meet the patient needs [28, 29].

We aimed to explore the expression of needs among patients with IE during hospitalisation and the first months after discharge.

## Methods

### Design

The study had an explorative inductive design using interviews and qualitative content analysis, according to Graneheim and Lundman [30]. The method describes phenomena systematically by structuring and interpreting transcribed verbal data [31, 32]. Variations in experiences are identified and classified at different abstraction levels. Through the interpretation process, new knowledge can emerge [31]. The study is reported in concordance with the consolidated criteria for reporting qualitative research (COREQ) 32-item checklist to ensure trustworthiness [33].

### Setting

The study was conducted in a region in Sweden with one primary and one secondary hospital. The care for patients with IE in the region is most often performed in the department of infectious diseases but sometimes also in internal medicine departments. According to national guidelines [34], the in-hospital time for treatment is between two and six weeks. Hospitals employ daily infectious disease consultants. Patients are referred to a tertiary hospital in a neighbouring region if they need cardiac surgery. A follow-up visit to an infectious disease specialist occurs 1–3 months post-discharge.

### Participants

Patients over 18 years of age with recent or ongoing in-hospital care due to IE were included in the study from 11th February 2022 to 23rd March 2023. During the period, a total of 46 patients with diagnoses of definite IE were registered in the region. Of these, 32 patients were asked for inclusion. Non-inclusions were mainly due to cognitive deficiencies such as dementia or death during hospitalisation. Eventually, 22 patients accepted the invitation, one withdrew consent, and one died before the interview, meaning that a total of 20 patients participated in this study. All 20 were mobilised and at home at the point of the interview. The baseline data of the participants are shown in Table 1.

### Data collection

The participants were interviewed a median of 112 days (67–221) after hospitalisation either in a neutral premise close to the ward (n = 6), at home (n = 5) or by telephone (n = 9). In three of the interviews, the spouses listened in on the conversation and added information. The semi-structured interviews focused on patients' expressed needs during hospitalisation due to IE, asking about their experiences of IE and the experiences of the care they had been given. The interviewers (HL and IL) were not involved in the care of the study patients. HL is a PhD student and consultant in infectious diseases, and IL is an experienced qualitative researcher and nurse specialising in the field of rheumatology. The interviews ranged from 24 to 107 minutes (median 70 minutes), were digitally recorded and transcribed verbatim.

### Data analysis

According to qualitative content analysis [31], the entire text was read several times by HL to better comprehend the dataset and gain a sense of the whole. Meanings relevant to the objective were identified and extracted, preserving the context. This resulted in 1635 meaning units. These were condensed and then abstracted and coded. The analysis process remained close to the text and codes were compared based on differences and similarities and grouped into eight subcategories and two categories reflecting the central message of the interviews; the manifest

**Table 1. Characteristics of the study population.** Sociodemographic and clinical data of the 20 study patients with IE.

| | | |
|---|---|---|
| Age | Years (median, range) | 74 (45–86) |
| Sex | Women | 6 |
| | Men | 14 |
| Highest level of education | Elementary school | 9 |
| | High school | 3 |
| | College | 8 |
| Working | | 3 |
| Religion or worldview | Secular | 12 |
| | Christian | 4 |
| | Atheist | 1 |
| | Other | 3 |
| Marital status | Married or common-law spouse | 14 |
| | Widow/widower | 5 |
| | Single | 1 |
| Environment | City | 7 |
| | Village | 6 |
| | Rural | 7 |
| Swedish born | | 18 |
| TEE performed | | 20 |
| Predisposing heart condition | Valve prosthesis | 5 |
| | Pacemaker | 2 |
| Thoracic procedure during hospitalisation | Thoracotomy | 4 |
| | Extraction of pacemaker | 2 |
| Hospitalisation | Days (median, range) | 28 (12–43) |
| Healthcare setting | Department of infectious diseases | 17 |
| | Department of medicine | 3 |
| Outpatient care | | 5[a] |
| | POET | 2 |
| | OPAT[1] | 4 |
| Microbiology | *S. aureus* | 4 |
| | NHBS | 7 |
| | *E. faecalis* | 4 |
| | Other[b] | 5 |
| Location for interview | Home | 5 |
| | Telephone | 9 |
| Time from hospitalisation to interview | Days (median, range) | 112(67–221) |

Abbreviations: TEE—transoesophageal echocardiography, NHBS- non-haemolytic β-streptococci, POET—Partial Oral Endocarditis Treatment, OPAT—Outpatient Antimicrobial Treatment

[a]One patient had OPAT and POET consecutively

[b]Other—*Staphylococcus capitis*, *Staphylococcus lugdunensis*, *Cutibacterium+Bartonella*, *Gemella morbillorum*, *Streptococcus dysgalactiae*

content (what the text says), and an overall theme reflecting the latent interpretation (the meaning of the text) (Table 2). HL performed the data analysis; the last author (IL), who has experience in qualitative methods, acted as a co-assessor, and continuous discussions were held until a consensus was reached and the whole research group confirmed the analysis. A healthcare professional with personal experience in IE contributed reflections on the analysis,

**Table 2. Analytical steps.** Examples of the analytical steps from meaning units to theme.

| Meaning units | Codes | Subcategory | Category | Theme |
|---|---|---|---|---|
| The shots and all the injections, they were hard towards the end, the antibiotics pained my body Participant no. 19 | Painful antibiotics | Need for treatment and follow-up | Needs for medical excellence | Needs as a spectrum between treating the disease and meeting the person with the illness |
| It was very unnecessary because I think that as soon as you have a fever for a week, you should go to the doctor and demand a proper examination. Participant no. 17 | Difficulties in diagnostics led to delay | Need for diagnosis and attention to complications | | |
| But if you lie here for a week then. . . I think that you should get your own room. Participant no. 18 | Respect for the person during a long hospitalisation | Need to be seen as unique | Needs of person-centred care | |
| I got information the whole time. Participant no. 1 | Need repetition of the information | Need for knowledge | | |

securing patient perspectives. The computer program NVivo 11 (QSR International Pty Ltd, 2016) organised the data material.

### Ethical considerations

Ethics approval for the study was obtained from the Swedish Ethical Review Authority (2021-06641-01), and the study was conducted in accordance with the Helsinki Declaration [35]. Participants received written study information on the ward from a healthcare professional or by mail after discharge. Up to two reminders were sent by mail, and one patient was informed by a medical interpreter by phone. Participants were informed that participation in the study was voluntary, and they could withdraw whenever they wanted without explanation or consequences. Information was given that confidentiality, integrity, and identity would be protected, and the results would be reported only on a group level, exemplified by short citations. No financial compensation was given. All participants gave written consent.

## Results

The expressed needs of patients suffering from IE resulted in the overall theme, a spectrum between treating the disease and meeting the person with the illness. The needs encompassed a dual focus on both medical excellence and person-centred care, (Table 3). Medical excellence was needed to optimally treat, supervise, and offer follow-up on this rare and severe disease; patients longed to come home, and there were issues of reliability in the healthcare system. Person-centred care was requested, including individualised information leading to knowledge, reorientation, the beginning of health restoration, and being met as a unique person.

### Needs for medical excellence

Patients expressed needs for medical excellence during and after hospitalisation due to the special features of IE as a rare disease that is difficult to diagnose, leads to severe wasting, and has

**Table 3. Result.** The overall theme, categories, and subcategories describing the expressed needs of patients suffering from IE during and after hospitalisation.

| Overall theme | Needs as a spectrum between treating the disease and meeting the person with the illness | |
|---|---|---|
| Category | Needs for medical excellence | Needs of person-centred care |
| Subcategory | Need for diagnosis and attention to complications | Need for knowledge |
| | Need for treatment and follow-up | Need for reorientation |
| | Need for prompt homecoming | Need for health restoration |
| | Need for reliable healthcare | Need to be seen as unique |

consequences due to long-term intravenous treatment. Patients expressed a need for correct treatment and follow-up, but also for a return to normal life at home as soon as possible. Patients feared nonconformance and adverse events and expressed a need for reliable healthcare.

**Need for a diagnosis and attention to complications—it takes knowhow.**   IE was an unknown disease for patients, and the experience of IE was elusive. The suspicion of IE was initially delayed in primary care but then led to disease-specific investigations that were technically demanding and different from common infectious diseases. Patients described problems with heart failure, complications after surgery, and adverse drug reactions. Patients expressed gratitude and security for medical excellence in the supervision of treatment and complications.

> They take it seriously and. . .do what is needed and what possibly would have been needed to be done, which in this case was a possible surgery. Participant no. 6

Different descriptions of experiences from treatment in the infectious diseases ward and in the internal medicine wards were offered regarding dissatisfaction with information patients had been given on IE and skills in the caring team in the internal medicine wards.

**Need for treatment and follow-up—experts formulating the plan.**   Patients described relief in having a treatable infectious disease but were shocked by the long treatment time in the hospital. Intravenous antibiotics were seen as necessary but disturbing. The effect of the intravenous therapy was experienced as elusive, and receiving antibiotics was often not even perceived as a treatment. The administration of the drugs fragmented patients' nights and days, but this was generally found to be acceptable. Some patients had substantial side effects from the antibiotics, and a painful aspect of the care was the continuous need for venous access. A central venous line was perceived as a relief by the patients.

> My arms were black and blue and totally full of puncture wounds, until I refused. They weren't allowed to stick me anymore and then they put a CICC on me and it became better. Participant no. 10

Patients with a shortened hospitalisation followed up by outpatient treatment expressed their gratitude for the medical excellence of this arrangement. However, they encountered logistical challenges when seeking assistance from primary care or home care facilities; issues which would have been familiar to the specialist unit. Another obstacle was a delay in obtaining oral antibiotics due to backorders at pharmacies.

> I know she (the physician) was doing as much as possible for me to come home. She tried to get me into treatment at an infirmary close by, but she wasn't able to. And then she got tablets for me which allowed me to come home within a reasonable time. Participant no. 12

After leaving the hospital, patients felt confident about knowing whom to contact in case of inquiries or if they felt unwell. The planned appointment at the infectious disease department was seen as important to understand and cope with residual low-key symptoms. While some patients had ongoing evaluations, others had routine visits that did not provide additional value for them.

> How are you doing now? I'm better. Good. And then everything was OK. It wasn't negative, but it also wasn't something that gave me an aha moment. Participant no. 8

**Need for a prompt homecoming—the urge for a normal life.** Regarding hospitalisation, patients were suddenly removed from their everyday lives by the disease, and they longed for home. The hospitalisation was experienced as long, boring, and distressing. It was seen as important when healthcare opened up to normalise life as much as possible. Examples included visits from friends and relatives, the use of digital contact venues, and gaining control over their economy. During the COVID-19 pandemic, different levels of non-permission for visitors were at hand, and those with limited visits missed them and regretted it. Digital possibilities for communication made home and everyday life come closer. Patients contacted friends and relatives and friends, and patients who were still working kept in touch with their jobs digitally.

> It was the most important. I was sitting there alone with absolutely nobody to talk to. When my husband and son came, I felt safe, I had someone to talk to. I think it has to be completely fine for them to be allowed to come. Participant no. 21

Although patients accepted shared rooms, they expressed a desire for more privacy. Having a separate room would have made them feel more at ease, secure, and relaxed, and they believed that recovery would have been faster without disturbing neighbours. Patients reported that the longer the hospital stay, the greater the need for privacy became apparent. Patients described various strategies to address the challenges of limited privacy, such as using the bed as a private place, using other areas within the ward, and using earplugs. Both male and female patients expressed issues of insecurity about mixed-gender rooms. Those with experiences of isolation care missed their freedom of movement, while others enjoyed the options to temporarily leave the room or ward.

> I think there is too much to take into consideration when sharing room with someone when you don't feel good. But 14 days is fine, you can do that. That's my opinion. Participant no. 18

To cope with boredom on the ward, patients engaged in self-initiated activities, such as watching TV, reading, playing cards, and taking walks. Healthcare lacked recreational options, offering only occasional short trips for investigations. Some patients—especially those living far away—expressed a desire for care closer to or within their home, since the treatment mainly involved antibiotics. On the other hand, they expressed the need for specialised care and medical excellence. Maintaining a normal family life was highly valued. Frequent outpatient contacts with physician and treatment controls provided reassurance and a sense of security during the treatment. The burden of responsibility on relatives was described as surprisingly heavy, and this made them hesitate about returning home too quickly in the future.

> If I had been able to come home instead and lie there at home. . .But not if there were any risks with it, then I would probably have passed. It was important that I had healthcare. Participant no. 7

**Need for reliable healthcare—wishing for transparency and security.** Patients expressed a need for medical excellence and security in the healthcare system and the long treatment period of hospitalisation increased the risks of adverse effects. Patients experienced delays in diagnosis, malpractice, and other shortcomings in healthcare, and there were experiences of adverse drug reactions. Patients' dependency on the care, and inequality in care was

exemplified by experiences of care failing to cope with patients' complex symptomatology, and in situations of being treated as an object of care instead of a unique person.

> They gave me a little plastic jar with a lid with medicines. Here are your morning medicines. . .It was strange to me that I was expected to just heave in medicines that I didn't know why I was taking. Participant no. 14

Patients explained the unreliable healthcare as due to a lack of time for meetings between patients and healthcare professionals. Patients reported that changes were made in treatment or investigations without any previous discussions, patients were suddenly moved from one ward to another, and meetings with physicians were unpredictable. There were negative experiences of healthcare professionals talking about the patient more than to the patient when waking up from unconsciousness.

> They should have time to treat every patient. . .It feels like there are shortages when it comes to the staff, maybe they don't have enough time for each patient. Participant no. 11

Transparency and responsibility from reliable healthcare were requested, but patients were unsure whether they could trust the healthcare system. Healthcare was viewed as hierarchical, and probably negatively biased towards patients' opinions. This negatively affected trust, but patients still tried to kindly interpret shortcomings in their care.

> I have a great trust in healthcare but sometimes you lose that trust, but that is because you get sick and feel bad and then you want everything to go quickly and you want to be seen, and here I am. But that is something you need to work on by yourself, but I still trust the healthcare system, I can come to the hospital by ambulance I can come to a doctor and talk to them even if they can't straighten things out. Participant no. 19

Respect for the patient's autonomy was described as important and connected to person-centred treatment. Confidentiality was seen as threatened, but this was not disturbing, since having IE was not regarded as a big secret. More worries were expressed about the integrity of their fellow roommates than their own, and patients expressed feeling responsible for keeping their roommate's secrecy.

### Needs for person-centred care

Patients expressed the importance of being met and listened to, that is, needs of person-centred care. Receiving personalised information on IE, increased knowledge of the disease, and understanding the treatment were prerequisites to starting the process of adaptation and reorientation towards health restoration. When healthcare professionals met the patient as a unique individual, this information and care, together with the professionals' experience of the special features of IE, led to satisfaction with care and a sense of security.

**Need for knowledge—never heard of this disease.** Since IE was unknown to the patients, they had a need for knowledge and requested customised and personalised information to understand the disease and expressed the need to return to the given information. The fact that IE affected the heart made a difference compared to other infections and was experienced as more threatening. The ongoing dialogue with repeated information from the physician and other healthcare professionals was important, and written information was seen as objective, instructive, and reassuring. Person-centred information on findings, stages of investigations, disease outcomes signalled security, and a lack of this information led to anxiety and a loss of

trust. Messages from healthcare professionals were sometimes hard to interpret, especially if they were given while patients were under stress and when healthcare professionals were unfamiliar with IE. Patients asked for prognostic information on IE in their personal lives after the follow-up visit and described an absence of information and a need for knowledge of long-term low-key symptoms.

> I haven't gotten very much information, but rather it's been more about how to get rid of the bacteria in the blood and aftercare in that way. Participant no. 19

Disabilities in hearing or vision, cognitive impairment, and language difficulties represented hindrances for sharing information between patients and personnel. A lack of information obliged the patients to rely on intimidating information from the internet. There were examples of patients not having their diagnosis explained to them, but only mentioned in their medical record, which was perceived as unstructured and containing fallacies. Patients cared for in specialised units described more person-centred care and satisfaction with the information they had been given.

> I actually didn't know that it was called that way (endocarditis) when they said that I had an infection because they never explained it to me. . .I think I met 10 doctors who all said different things. Participant no. 10

Patients informed their network, and some were content with this, but others missed healthcare support to their relatives. A lack of information during critical periods caused worries for both those at home and for the patient. Patients experiencing acute surgery perceived a lack of person-centred information in the gaps between care units.

**Need for reorientation—reacting to and processing what has happened.** The diagnosis of IE and other disturbing information was deeply concerning, and patients felt the threat to health and life from the disease and its complications. Information about the long hospitalisation forced them to adjust their immediate future plans and social life. The skill of professionals in communicating with patients and maintaining a person-centred approach made a great difference to patient wellbeing. The diagnosis, perceived frailty, and cognitive impairment led to processes of spiritual and psychological reorientation. Spiritual needs concerned reflections on life and death, hope and doubts, guilt and shame—and, for some patients, a deep feeling of gratitude.

> Super scared that my heart won't manage because somewhere in the back of my mind I'm thinking why did they put in a pacemaker, that I have. When I took it out in 2006 they said I wouldn't survive and it's 2023 now. But at the same time, I still have their words in the back of my mind, it's experts we're talking about. Participant no. 19

Facing a life-threatening situation prompted preparations and plans for their own funeral and the future life of their spouse. Patients hoped to regain strength but, at the same time, doubted they would. In cases where there was a long delay in seeking healthcare, both patients and their relatives felt guilt. Reinforced bonding with their families, joy, and even happiness in being alive and feeling better, were also reported. Those who had faith referred to it as a consolation and source of security, while others described confidence in their self-assurance. In the process of accepting and reorienting, patients emphasised the importance of person-centred care and of dialogue, especially with family and friends, in coping with the disease and its aftermath. There were patients who experienced anxiety and signs of depression during and after

the disease, hampering their reorientation. These symptoms were new and unwelcome to them, leading to limitations in their everyday lives.

> A little depressed, yes. I can feel that and it's a strange feeling. I have very rarely been depressed. Participant no. 9

Person-centred contact with a social worker was important to some patients, and contact was sometimes pursued after discharge. Patients without an offer of psychological or spiritual support did not regret this. Information on prophylactic measures against bacteraemia led to anxious thoughts, and every manipulation of the teeth or a skin scratch was a reminder that IE might arise again.

> Now I'm very scared, was actually at the dentist the other day to check. Participant no. 2

After discharge, reorientation was affected by impairments in cognition, memory, and hearing related to the disease or to treatment. This was not obvious on the ward and was experienced as deeply concerning and affected relations at home. Patients hoped that time would heal this.

> There was nothing wrong with either my arms or legs but there was something wrong in the head. I don't feel like before. When it comes to those things, those cognitive things. Participant no. 14

**Need for health restoration—a long journey.** The physical aspects of low-key symptoms, such as unquenchable fatigue, loss of appetite, weight loss of up to fifteen kilograms, and severe muscle wasting with changes in habitus seen as empty skinfolds, were described. These symptoms were substantial, resulting from contracting IE and subsequent convalescence, and health restoration was needed.

> I have a feeling that I'm still living with this disease. That's how it feels. Participant no. 21

Eating and enjoying food was difficult for patients, and sometimes oral intake was almost impossible during hospitalisation due to changes in the sense of taste. Personalised treatment with special offerings and nutritional drinks were appreciated, and opinions on the food served in the hospital differed. Nutrition was seen as important to regain health but could be overwhelmingly hard to achieve.

> I couldn't eat so much when normally food tastes so good. Coffee I didn't even touch for all those weeks, I couldn't drink coffee, normally I'm such a coffee lady. Participant no. 5

Frailty threatened safe homecoming given the loss of mobility functions and cognitive effects due to IE, and planning was important for restoration of the social part of health. There were examples of private adjustments as well as help from a social planning team. Patients did not experience low-key symptoms being addressed as a routine, and meetings with other healthcare professionals were not experienced as part of a structured plan. The few encounters reported with physiotherapists, occupational therapists, or dietitians to restore health were occasional in most cases. The possibility of person-centred healthcare support and professional rehabilitation efforts for long-term health was not known to patients, and therefore was not anticipated by patients.

(About exercising) But we hadn't talked about it at all. Maybe they saw that I was walking so much that that was enough. Participant no. 9

Months after discharge, patients described the need for health restoration even though they felt better. Tiredness and weight loss still meant low physical capacity, and while they hoped for improvements, catch-up was slow.

**Need to be seen as unique—personalisation increasingly important over time.** The long duration of hospitalisation increased the need for person-centred care and to be seen as a partner; unique, listened to, and well-treated by professionals. Meeting the needs most central to patients was increasingly important, and giving time, patience, respect, and showing competence led feelings of security, confidence, and empowerment. Patients acknowledged the work environment and appreciated the continuity of care from the multiprofessional team, giving a feeling of resemblance, predictability, and control in a vulnerable situation. The capacity of the healthcare professionals to deal with unanticipated needs was trusted, and their compassion and good humour were admired.

I trusted them, they were constantly in taking tests and blood and all those kinds of things. They were professionals you thought, they clearly knew these things. Participant no. 2

Continuity of care led to relationships based on mutual trust, and when knowledge of one another led to person-centred care, this was perceived as affirming. Individualisation of practical things was seen as precious. Respecting the patient as part of a social context led to participation in care by transmitting knowledge to the patient's relatives and the ability to influence day-to-day life. Patients experienced participation when healthcare professionals listened to their previous experiences and felt they were a part of the team fighting the disease. The medical decisions, however, were left to the experts.

I'm a layman, I shouldn't get involved with stuff like that. Participant no. 6

Generally, the patients described a feeling of safety and well-being in the wards when engaged healthcare professionals treated them with kindness—this was appreciated and experienced as being seen and treated as unique persons.

I think it's important, that everyone has been honest. Naturally if I'm secure that you are treated as you are, that's how it is. Participant no. 5

## Discussion

This study explores the expressed needs of patients suffering from IE during and after hospitalisation and shows the needs on a spectrum between treating the disease and meeting the person with the illness, thus between medical excellence and person-centred care. Our study highlights specific IE-related needs of medical excellence that are congruent with the ambitions of healthcare, such as diagnosing IE, giving the best treatment in the shortest possible time, and being attentive to complications. But patients also experienced struggles with healthcare concerning their safety during the long hospitalisation period, perceived frailty, and the consequences of a lacking person-centred approach.

Diagnosing and treating IE is a complex task [1, 19], and in our study, shortcomings in suspecting IE in primary care were revealed. An increased awareness of the importance of taking blood cultures may be a way to shorten the delay in diagnosing IE. In our study, the long time

before diagnosis and the long hospitalisation led to experiences of different adverse events, leading to doubts about the healthcare system. When reliable healthcare is lacking, trust diminishes, and anxiety increases [36]. Distress about care is ameliorated when healthcare professionals show competence in medical, physical, psychological, spiritual, and social matters. The professionals in our study, devoted to the special characteristics of IE, including low-key symptoms [19, 37–40] offers a basis for tailoring care to the patient and thus, ease distress; for example, by securing a central line early or arranging a switch to outpatient care [41]. Our study exhibits the need to normalise life, and most importantly for patients, to return home sooner, since a long hospitalisation period has a large impact on the life of the patient and their relatives [42, 43]. In our study there were some patients who were discharged with outpatient care and others who expressed a need for a prompt discharge but were not offered this. This shows that the new concepts of treatment [1, 34] are now challenging the dogma of treatment but the new framework of IE care, defined by guidelines, are not yet fully implemented [21, 44]. Treatment must be customised to each patient due to different factors of the disease, the patient, and the healthcare system, and the implementation of Partial Oral Treatment of Endocarditis (POET) [19, 21, 45] and Outpatient Parental Antimicrobial Treatment (OPAT) [44, 46] emphasises this need. Experiences from our study revealed logistical challenges in the switch to outpatient care, demanding skills from a multiprofessional team [19]. Patients who had outpatient treatment were satisfied with this normalisation of life and felt safe after discharge, but there were also issues of responsibility [47].

The needs of cure and knowledge of IE in our study were met in the specialised unit, and the competence of the team was valued. Patients cared for in general units expressed less satisfaction, mainly due to information issues and lack of person-centred treatment. Due to these challenges, from diagnostics to treatments, there is a recommendation to centralise the care of IE [7, 19]. The ESC 2023 guidelines point to the importance of specialist competence and resources in IE care to safeguard equality and quality [19].

Our results show a great and protracted impact on patients' lives leading to consequences for energy and mental health. Previous studies on IE have indicated longstanding worsened health physically and mentally [8, 48]. The long-term results from the POET study showed the cohort randomised to oral switch and quicker discharge have a more favourable outcome over time [41], indicating the importance of normalising life as soon as possible to enhance the restoration of health and quality of life [13, 42, 49]. Since the regain of health after IE progresses more slowly compared to other cardiac ailments [8, 50], this could affect secondary costs of readmission [8, 51]. However, there is still a gap in knowledge concerning which patients are suitable for outpatient care or shortened treatment, especially since the group of patients with complicated IE is increasing [19, 45, 52].

Guidelines on how to diagnose and treat IE are well known [1, 19], but guidelines on person-centred care on partnership and patient narratives leading to personal health plans [53] are less well-known and implemented. Hospitalisation is associated with risks of iatrogenic harm [54], issues of premises and everyday routines in a ward are important, and isolation results in negative patient experiences [36, 55]. Coping with situations of inequality is described as distressful in our study, feeling like an object more than a subject in healthcare and the experience of sharing rooms with others becomes tougher over time. The examples of distress in care underline the importance of medical excellence and person-centred care [24, 36, 55]. Our result shows the importance of offering information about the condition, leading to knowledge as a cornerstone for patients in accepting this previously unknown disease and its consequences for the future [19, 24]. The expressed needs for personalised and repeated information to increase knowledge are in line with previous reports of patient experiences in health care [42, 46, 56]. Information gave patients the ability to participate and the patient felt

respected. Productive meetings with healthcare professionals led to feelings of security, and being seen and known as unique, but if these interpersonal meetings were hampered by stress and limited resources, they impaired the quality of care. This result describes central parts of person-centred care, which starts in the patient narrative, works to build a partnership in shared decision-making, and is eventually safeguarded in documentation of the narrative to ensure transparency of the patient–healthcare interplay [53].

In our study, patients are satisfied with the in-hospital aspects of IE but express a lack of knowledge as to how the disease impacts the individual over time. This is a question still lingering after hospitalisation and follow-up visits, hampering the reorientation and recovery of the patients; this is also seen in previous studies [17, 42]. The extended time for health resolution is worrying for patients, pointing to the need for enhancements in the treatment and observation of persistent symptoms of wasting, memory disturbances, anxiety, and depression [8, 51]. Our results point to a need to address the restoration of health since it is less well met compared to more acute needs, even in the specialist department. Plans for rehabilitation and the evaluation of long-standing symptoms are not routine, and these low-key symptoms are veiled behind more acute and threatening symptoms. The study shows few encounters with other healthcare professionals, either during or after hospitalisation. Low-key symptoms are not recognised by patients as a need that can be addressed by healthcare [43, 57, 58], which leads to silent suffering. There is currently a lack of knowledge on the trajectory of IE and other severe bacterial infections [8, 59, 60]; a randomised controlled trial on multimodal rehabilitation for patients with IE was inconclusive [13], and to date, it is unknown whether interventions can mitigate the physical and mental health of patients with IE [50, 57, 61]. Follow-up visits are appreciated, but are unstructured and risk missing anxiety, depression, and cognitive effects, and can even increase worries about everyday life depending on the way prophylactic information is delivered [18, 62]. Our findings support the need for improvements in healthcare to aid patient recovery. Challenges and possibilities for healthcare in meeting the needs for prompt homecoming and health restoration expressed in this study concern competence provision and resource transfer to outpatient care with continued safety and person-centred care [19–21]. The multiprofessional team needs to extend the complex and intertwined processes of medical excellence and person-centred care to the outpatient level when care changes. To succeed in this implementation, healthcare organisations need to develop new strategies [19, 53].

Further research is needed to increase knowledge and improve management of the low-key symptoms of IE. The patient perspective in the field of infectious diseases is identified as a gap in knowledge [19, 29, 63, 64], and new models of multiprofessional care [65, 66], rehabilitation [50, 57], and screening [67] of reasons for delays in health progress, as well as person-centred care with shared decision-making [24], and the evaluation of outcomes other than pure survival are necessary [28, 61, 68–71].

## Strengths and limitations

In qualitative research, trustworthiness is defined according to the four criteria of credibility, dependability, confirmability, and transferability [30, 31]. To explore the variation and diversity of experiences, qualitative content analysis was chosen. The strengths of our study are the diverse backgrounds of the patients. Inclusion from a regional population make the results transferable to similar settings. Based on a broad aim, dense specificity, strong dialogue, and cross-case analysis, the inclusion of 20 patients was judged sufficient to reach information power. The point of study end was guided by the information yielded during the study [72]. The interview texts were rich in meaning and variation, strengthening their credibility.

Interpretation and sorting of the content can be influenced by the researcher's precepts and ability to validate. Accounting for the interviewer's preconceptions about IE, the stepwise analysis, having an experienced qualitative researcher (IL) as a co-assessor as well as confirmation of the resulting categories and subcategories with a medical expert and leader of an endocarditis team (MR) and a patient representative increased the dependability. Limitations of the study consisted of the absence of intravenous drug users and few patients born outside of Sweden. The rich content was deemed sufficient, but the patients who refused inclusion might have contributed other aspects.

## Conclusions

This explorative study of patients suffering from IE during and after hospitalisation showed eight specific important areas of improvements to pin-point in meeting patient's needs. The needs were described as a spectrum between treating the disease and meeting the person with the illness, or between medical excellence and person-centred care. Patients with IE require care in specialised units where personalised information, together with the professionals' experience of the special features of IE lead to knowledge of the disease. This starts the process of reorientation towards health restoration. Symptoms of fatigue, wasting, and cognitive and mental distress are longstanding and are often neglected in healthcare. Meeting the patient as a unique individual leads to satisfaction and a sense of security. Our study is an important step forward and highlights the need for improvements in healthcare to guide patients towards recovery after IE. Early discharge is important and demands the implementation of new treatment algorithms leading to changes in healthcare setting. The complex person-centred teamwork of the specialised unit needs to be extended to the outpatient level with preserved safety. More knowledge is needed on the disease trajectory after discharge, and the possibility of impinging the course of remediation, as well as domains of rehabilitation and the role of screening for delayed restoration to health. Multiprofessional individualizing care as directed by this groundwork, are therefore fields for future qualitative and quantitative studies.

## Acknowledgments

We would like to thank Ms. Sofie Kostic for valuable brainstorming before constructing the interview guide, Ms. Malou Leksell for her work with the inclusion of patients in this study, all nurses who distributed the information, and Ms. Disa Lundstroem for transcribing interviews, all working at the Department of Infectious Diseases, Halmstad. Thanks to Dr. Majed Faleh for enabling purposeful inclusion and Ms. Elizabeth Thomas for interpretation of citations and valuable input in the English nuances of the overall theme. From our hearts, we also thank the research participants for generously sharing their experiences.

## Author Contributions

**Conceptualization:** Helena Lindberg, Magnus Rasmussen, Ingrid Larsson.

**Data curation:** Helena Lindberg, Ingrid Larsson.

**Formal analysis:** Johan Vaktnäs, Magnus Rasmussen, Ingrid Larsson.

**Investigation:** Helena Lindberg, Ingrid Larsson.

**Methodology:** Helena Lindberg, Ingrid Larsson.

**Project administration:** Helena Lindberg.

**Resources:** Ingrid Larsson.

**Supervision:** Johan Vaktnäs, Magnus Rasmussen, Ingrid Larsson.

**Validation:** Johan Vaktnäs, Magnus Rasmussen, Ingrid Larsson.

**Writing – original draft:** Helena Lindberg.

**Writing – review & editing:** Helena Lindberg, Magnus Rasmussen, Ingrid Larsson.

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
