## [Decision Letter · Decision Letter 0]

5 Jun 2024

PONE-D-24-10353Treating the disease and meeting the person with the illness-patient perspectives of needs during infective endocarditis, a qualitative studyPLOS ONE

Dear Dr. Lindberg,

Thank you for submitting your manuscript to PLOS ONE. After careful consideration, we feel that it has merit but does not fully meet PLOS ONE’s publication criteria as it currently stands. Therefore, we invite you to submit a revised version of the manuscript that addresses the points raised during the review process.

**ACADEMIC EDITOR: I agree with the reviewers that the manuscript is well-written. The study applies to PLOS ONE readership. As noted by the reviewers, the table numbering must be corrected for the resubmitted manuscript.   **==============================

We look forward to receiving your revised manuscript.

Kind regards,

Che Matthew Harris

Academic Editor

PLOS ONE

Journal Requirements:

2. In the online submission form, you indicated that the datasets analysed during the current study are not publicly available due to the

integrity of the interviewees but are available from the corresponding author upon

reasonable request

Reviewers' comments:

Reviewer's Responses to Questions

**Comments to the Author**

1. Is the manuscript technically sound, and do the data support the conclusions?

Reviewer #1: Yes

Reviewer #2: Yes

2. Has the statistical analysis been performed appropriately and rigorously? 

Reviewer #1: N/A

Reviewer #2: N/A

3. Have the authors made all data underlying the findings in their manuscript fully available?

Reviewer #1: No

Reviewer #2: Yes

4. Is the manuscript presented in an intelligible fashion and written in standard English?

Reviewer #1: Yes

Reviewer #2: Yes

5. Review Comments to the Author

Reviewer #1: This is a study on infective endocarditis based on the perception of the patient who has made a series of considerations on general and technical aspects of his disease. The structure of the article is adequate in that it correctly addresses the state of the problem in the introduction, analyzes what has been published previously and the significance of the study in the discussion, and describes the possible limitations.

In this study, the authors delve into the feelings, doubts, expectations, demands, and desires of patients after being diagnosed with such a serious infection that usually requires prolonged hospitalization or surgery as infective endocarditis.

This study represents an opportunity to learn about all the subjective aspects surrounding this disease and its content may be of interest to clinicians.

As a result of the study, it is clear that patients need to perceive a deep knowledge of this pathology on the part of the physicians responsible for each case. In this sense, it seems that patients treated by specialists in infectious diseases are more satisfied than those treated by more general practitioners such as specialists in internal medicine. However, I believe that the time spent with patients, the patient's perception of diagnostic accuracy and the quality of care may not be so much related to the specialty of the treating physician, but rather to professional and personal aspects of each physician.

The demand for information regarding the impact on cardiac function, general physical and psychological deterioration that this infection produces in most patients has also been evident in this research.This article also allows physicians who treat infections in general and infective endocarditis in particular to reflect on patients' fears, anxieties and expectations of recovery. It is, therefore, an interesting investigation, but it does not analyze quantitative data on the clinical course of this serious infection.

Among the negative aspects of the work, the small number of participating patients and its long length stand out. Despite the size of the manuscript, we can point out that it is easy to read and that important aspects of the patient-physician relationship that are not usually addressed in conventional clinical practice are discovered.

As specific aspects, we highlight that Table 2 is doubly identified,

This article would also have a place in journals focused on health care, nursing journals and journals of a more general or informative nature. There is no clear research question or specific objectives, but rather a description of how a patient lives and feels when diagnosed with infective endocarditis.

Reviewer #2: Comments to the authors

Your paper is an atypical but very interesting paper on how the patients experience an episode of infective endocarditis. From the interview of 20 patients, you individualized 2 main axes of comments / patients’ needs: medical excellence and patient-centred care. You also identified 4 subsections for each.

This paper is very interesting, because it describes exactly the issues we encounter in our daily practice. Because IE is a complicated disease, we need to be highly specialised and reliable to manage patients the best as possible (for all steps of patient’s management: diagnosis, treatment and follow-up). Because IE medical therapy is long, we need to organise early return at home but with a reliable follow-up also during this period of treatment and during the early post therapy period. Because IE is a rare disease, we need to explain it clearly and comprehensively to patients and families and we have to take enough time for each of them and to accompany them during hospitalisation and after (patients’ need for knowledge, reorientation, health restoration and individualization)!

I have very few comments as your paper is really easy to read and well written.

The main point to be improved is the abstract as it does not clearly reflect the paper. It was the first thing I read, and I found it quite hard to understand… In the abstract result section, you should clearly identify each category, with its subcategories, reported on the same manner as in your paper (in the current form of the abstract, some ideas are separated by semicolon, others are complete sentences, it is really not clear! for instance, the first category is the need for medical excellence ; but when reading your abstract, I’m not able to identify the subcategories of this category…). You should use the same terminology as in table 3 (and not Table2), or as in the first paragraph of results, which is easier to understand. In the abstract conclusion, there are too many ideas in the same sentence: patients want to be treated in specialised units, they want to be discharged early, and the third idea is around outpatient care… Cut your sentences, “one sentence for one idea!” as I’ve always been taught!

Details:

The introduction paragraph may probably be shortened, as details about IE risk factors or complications are probably out of topic here. Do not be too strict when reporting percentages (Cardiac surgery is performed in 30% of cases…) add around 30% or give ranges (30-50%).

You should perhaps probably add somewhere in your discussion this paper on adherence to oral hygiene following an episode of IE which clearly shows that depression and cognitive impairment impact the adherence to important healthcare following an IE (Celestin B Determinants of adherence to oral hygiene prophylaxis guidelines inpatients with previous infective endocarditis Arch Cardiovasc Dis. 2023 Apr;116(4):176-182)

You should perhaps also suggest the help of psychologists during IE therapy or among IE teams.

In the conclusion, you could perhaps underline that individualizing these 8 axes of potential improvement in the daily management of IE patients is a great achievement that can open the door to other qualitative studies evaluating the impact of subsequent changes of behaviour of caregivers, as you mention in your last sentence.

The second sentence of your conclusion is not clear, the last part beginning with ”start the process” is not properly linked to the beginning of the sentence…

6. PLOS authors have the option to publish the peer review history of their article (what does this mean?). If published, this will include your full peer review and any attached files.

Reviewer #1: **Yes: **Antonio Ramos-Martínez

Reviewer #2: **Yes: **Christine Selton-Suty

---

## [Author Response · Author response to Decision Letter 0]

13 Jul 2024

Response to Editor and Reviewers, Halmstad 240624

We have read Your conclusions and comments, and our work will benefit greatly from them. We are grateful for Your suggestions of improvement of our work and delighted that You found the subject important and our manuscript interesting.

We have revised the work according to Your suggestions. Comments and discussions are addressed below.

ACADEMIC EDITOR: I agree with the reviewers that the manuscript is well-written. The study applies to PLOS ONE readership. As noted by the reviewers, the table numbering must be corrected for the resubmitted manuscript. 

Answer: We have corrected the table numbering and adjusted the title and affiliations according to the PLOS ONE style templates.

2. In the online submission form, you indicated that the datasets analysed during the current study are not publicly available due to the

integrity of the interviewees but are available from the corresponding author upon

reasonable request

Answer: Our data consists of transcribed interviews in Swedish containing personal information. Sharing this pose risks of breaches in the patient’s integrity. The ethical approval was based on the confidentiality request by the Helsinki declaration. The study data were anonymized after the transcription and grouping, to secure this confidentiality during the process and publication and this plan of data handling was acknowledged by the Ethical approval authorities. Sharing these texts “raw”, i.e. before the anonymization and grouping step could therefore lead to identification of individual patients based on descriptions of personal circumstances since the group of patients is small and from a defined area in Sweden.

Answer: We have no Supporting information, only three tables in the text. We have adjusted the captions and the in-text citations of them.

Answer: We have reviewed the reference list and have not found any retracted papers. The list is unchanged.

Reviewer #1: This is a study on infective endocarditis based on the perception of the patient who has made a series of considerations on general and technical aspects of his disease. The structure of the article is adequate in that it correctly addresses the state of the problem in the introduction, analyzes what has been published previously and the significance of the study in the discussion, and describes the possible limitations.

Answer: Thank You for Your comments.

In this study, the authors delve into the feelings, doubts, expectations, demands, and desires of patients after being diagnosed with such a serious infection that usually requires prolonged hospitalization or surgery as infective endocarditis.

This study represents an opportunity to learn about all the subjective aspects surrounding this disease and its content may be of interest to clinicians.

Answer: We appreciate the affirmative evaluations of our aim to highlight the patient’s perspectives.

As a result of the study, it is clear that patients need to perceive a deep knowledge of this pathology on the part of the physicians responsible for each case. In this sense, it seems that patients treated by specialists in infectious diseases are more satisfied than those treated by more general practitioners such as specialists in internal medicine. However, I believe that the time spent with patients, the patient's perception of diagnostic accuracy and the quality of care may not be so much related to the specialty of the treating physician, but rather to professional and personal aspects of each physician.

Answer: We aimed to describe the importance of medical excellence of the professionals to be able to give a good technical and personal care. We agree that the name of the speciality is not the primary issue, and the care is organized differently in different settings. In our setting, patients with endocarditis are sometimes cared for in general internal medicine wards and, only in cases with suspected arrythmias, in a cardiology ward. These patients perceived a lack of engagement, knowledge and perhaps therefore a disinterest of their disease in wards with a lack of team-based competence where the diagnose was rare. We have changed from infectious disease unit to specialist ward or specialised unit in the discussion (p 23, line 508).

The demand for information regarding the impact on cardiac function, general physical and psychological deterioration that this infection produces in most patients has also been evident in this research.This article also allows physicians who treat infections in general and infective endocarditis in particular to reflect on patients' fears, anxieties and expectations of recovery. It is, therefore, an interesting investigation, but it does not analyze quantitative data on the clinical course of this serious infection.

Answer: We appreciate the attention of the importance of knowledge in this area. It is correct that the study does not analyse quantitative data since it has an inductive qualitative study design.

Among the negative aspects of the work, the small number of participating patients and its long length stand out. Despite the size of the manuscript, we can point out that it is easy to read and that important aspects of the patient-physician relationship that are not usually addressed in conventional clinical practice are discovered.

Answer: Since this is a qualitative study the aspect of numbers of participating patients is different in this kind of design and the data these 20 patients contributed with were rich. To clarify this, we have added a reference and comment on this on p. 23, lines 514-517. We are aware that the length of the paper can be a hindrance to read it, but we have done our best to keep it as short as possible and acknowledge the experience of it as easy to read. We have made minor shortenings of the discussion to meet the reviewers demands (several places in the manuscript).

As specific aspects, we highlight that Table 2 is doubly identified,

Answer: The tables are rectified.

This article would also have a place in journals focused on health care, nursing journals and journals of a more general or informative nature. There is no clear research question or specific objectives, but rather a description of how a patient lives and feels when diagnosed with infective endocarditis.

Answer: The research question is, due to the qualitative inductive design, open. We have revised the formulation of the research question to make it even clearer.

Reviewer #2: Comments to the authors

Your paper is an atypical but very interesting paper on how the patients experience an episode of infective endocarditis. From the interview of 20 patients, you individualized 2 main axes of comments / patients’ needs: medical excellence and patient-centred care. You also identified 4 subsections for each.

Answer: We are thankful for the appreciating summary.

This paper is very interesting, because it describes exactly the issues we encounter in our daily practice. Because IE is a complicated disease, we need to be highly specialised and reliable to manage patients the best as possible (for all steps of patient’s management: diagnosis, treatment and follow-up). Because IE medical therapy is long, we need to organise early return at home but with a reliable follow-up also during this period of treatment and during the early post therapy period. Because IE is a rare disease, we need to explain it clearly and comprehensively to patients and families and we have to take enough time for each of them and to accompany them during hospitalisation and after (patients’ need for knowledge, reorientation, health restoration and individualization)!

Answer: We share the same experience of this complex disease and the challenges in handling it person-centred as described. 

I have very few comments as your paper is really easy to read and well written.

The main point to be improved is the abstract as it does not clearly reflect the paper. It was the first thing I read, and I found it quite hard to understand… In the abstract result section, you should clearly identify each category, with its subcategories, reported on the same manner as in your paper (in the current form of the abstract, some ideas are separated by semicolon, others are complete sentences, it is really not clear! for instance, the first category is the need for medical excellence ; but when reading your abstract, I’m not able to identify the subcategories of this category…). You should use the same terminology as in table 3 (and not Table2), or as in the first paragraph of results, which is easier to understand. In the abstract conclusion, there are too many ideas in the same sentence: patients want to be treated in specialised units, they want to be discharged early, and the third idea is around outpatient care… Cut your sentences, “one sentence for one idea!” as I’ve always been taught!

Answer: We are thankful for the valuable critique of the abstract. We have revised it (p. 2, lines 33-48) according to the comments and hope it will be more unanimous with the paper.

Details:

The introduction paragraph may probably be shortened, as details about IE risk factors or complications are probably out of topic here. Do not be too strict when reporting percentages (Cardiac surgery is performed in 30% of cases…) add around 30% or give ranges (30-50%).

Answer: We have made shortenings and changes of the text in the introduction according to the suggestions on the details and complications of IE as well as the figures.

You should perhaps probably add somewhere in your discussion this paper on adherence to oral hygiene following an episode of IE which clearly shows that depression and cognitive impairment impact the adherence to important healthcare following an IE (Celestin B Determinants of adherence to oral hygiene prophylaxis guidelines inpatients with previous infective endocarditis Arch Cardiovasc Dis. 2023 Apr;116(4):176-182)

Answer: We have read the suggested paper with great interest and have included the paper among our references. 

You should perhaps also suggest the help of psychologists during IE therapy or among IE teams.

Answer: This is an important reflection. However, different settings work in different ways and we chose to not separate the professions needed in our texts. None mentioned is none forgotten. We totally agree on the importance of the team-based care to offer services for both the mental and spiritual needs of the patients.

In the conclusion, you could perhaps underline that individualizing these 8 axes of potential improvement in the daily management of IE patients is a great achievement that can open the door to other qualitative studies evaluating the impact of subsequent changes of behaviour of caregivers, as you mention in your last sentence.

Answer: We have underlined the importance and consequences of our results (p. 25, lines 553-554, 561, 564, 566-570).

The second sentence of your conclusion is not clear, the last part beginning with ”start the process” is not properly linked to the beginning of the sentence…

Answer: The sentence is corrected.

In the manuscript the study persons has also been changed to Participant no. X etc and an ackowledgement has been included (p. 25, lines 546-553). We hope You find the corrections acceptable and look forward to Your further evaluation of our paper.

Best regards

Helena Lindberg

PhD student Lund university

MD, Senior consultant, Department of Infectious diseases, Region Halland

---

## [Decision Letter · Decision Letter 1]

12 Aug 2024

Treating the disease and meeting the person with the illness - patient perspectives of needs during infective endocarditis, a qualitative study

PONE-D-24-10353R1

Dear Dr. Lindberg,

We’re pleased to inform you that your manuscript has been judged scientifically suitable for publication and will be formally accepted for publication once it meets all outstanding technical requirements.

Kind regards,

Che Matthew Harris

Academic Editor

PLOS ONE

Additional Editor Comments (optional):

Reviewers' comments:

Reviewer's Responses to Questions

**Comments to the Author**

1. If the authors have adequately addressed your comments raised in a previous round of review and you feel that this manuscript is now acceptable for publication, you may indicate that here to bypass the “Comments to the Author” section, enter your conflict of interest statement in the “Confidential to Editor” section, and submit your "Accept" recommendation.

Reviewer #1: All comments have been addressed

Reviewer #2: All comments have been addressed

2. Is the manuscript technically sound, and do the data support the conclusions?

Reviewer #1: Partly

Reviewer #2: Yes

3. Has the statistical analysis been performed appropriately and rigorously? 

Reviewer #1: Yes

Reviewer #2: Yes

4. Have the authors made all data underlying the findings in their manuscript fully available?

Reviewer #1: Yes

Reviewer #2: No

5. Is the manuscript presented in an intelligible fashion and written in standard English?

Reviewer #1: Yes

Reviewer #2: Yes

6. Review Comments to the Author

Reviewer #1: The authors have addressed all my recommendations

I think it could be publishednit its current form

Reviewer #2: (No Response)

7. PLOS authors have the option to publish the peer review history of their article (what does this mean?). If published, this will include your full peer review and any attached files.

Reviewer #1: **Yes: **Antonio Ramos-Martínez

Reviewer #2: No

---

## [Editor Report · Acceptance letter]

15 Aug 2024

PONE-D-24-10353R1 

PLOS ONE

Dear Dr. Lindberg, 

I'm pleased to inform you that your manuscript has been deemed suitable for publication in PLOS ONE. Congratulations! Your manuscript is now being handed over to our production team.

Kind regards, 

on behalf of

Dr. Che Matthew Harris 

Academic Editor

PLOS ONE